# Investigation of Advanced Robotized Polymer Sheet Incremental Forming Process

**DOI:** 10.3390/s21227459

**Published:** 2021-11-10

**Authors:** Vytautas Ostasevicius, Darius Eidukynas, Vytautas Jurenas, Ieva Paleviciute, Marius Gudauskis, Valdas Grigaliunas

**Affiliations:** Institute of Mechatronics, Kaunas University of Technology, Studentu Street 56, LT-51424 Kaunas, Lithuania; darius.eidukynas@ktu.lt (D.E.); vytautas.jurenas@ktu.lt (V.J.); ieva.paleviciute@ktu.edu (I.P.); marius.gudauskis@ktu.lt (M.G.); valdas.grigaliunas@ktu.lt (V.G.)

**Keywords:** incremental sheet forming, advanced manufacturing technology, force reduction, robotized manufacturing, thermal modelling

## Abstract

The aim of this work is to evaluate the possibility of inexpensively producing small-batch polymer sheet components using robotized single point incremental forming (SPIF) without backing plate support. An innovative method of thermal and ultrasound assisted deformation of a polymer sheet is proposed using a tool with a sphere mounted in a ring-shaped magnetic holder, the friction of which with the tool holder is reduced by ultrasound, and the heating is performed by a laser. The heated tool moving on the sheet surface locally increases the plasticity of the polyvinyl chloride (PVC) polymer in the contact zone with less deforming force does not reducing the stiffness of the polymer around the tool contact area and eliminating the need for a backing plate. The free 3D rotating ball also changes the slip of the tool on the surface of the polymer sheet by the rolling, thereby improving the surface quality of the product. The finite element method (FEM) allowed the virtual evaluation of the deformation parameters of the SPIF. Significant process parameters were found, and the behavior of the heated polymer sheet was determined.

## 1. Introduction

Complex geometric shapes and details of various purpose objects and figures are rapidly increasing nowadays in various fields of industry and everyday life. Despite the fact, that modern computer hardware and software greatly facilitate the creation of such objects/shapes during the design phase, the actual production of such objects is very complicated and expensive. For this reason, the material sheet single point incremental forming (SPIF), mainly made from metals, is being investigated and used nowadays. This machining method allows a significant reduction in time and cost. Among other things, the SPIF allows the production of objects/shapes of extremely complex geometric shape which would be very difficult or even impossible to produce by traditional methods.

The SPIF principal concept is to progressively enable localized deformations by moving a generic shaped tool along a tool path to accumulate the whole shape of the part [1,2,3,4]. The paper [1] provides an overview of concurrent research on emerging trends in thermoplastics-oriented SPIF processes. The paper [2] attempt to use polymers for the first time to develop low-cost small-batch and high-quality products. The experiments were performed with polyvinyl chloride (PVC). The forming tool was mounted in a spindle with a bearing, which means that the tool rotates freely. This made it possible to eliminate the 1D forced rotation of the tool and to prevent sliding friction, reducing tool wear and forming forces. The authors of the paper [3] compared five different polymers: polyoxymethylene (POM), polyethylene (PE), polyamide (PA), polyvinyl chloride (PVC), and polycarbonate (PC), which are listed in descending order of crystallinity. It was found that increasing the tool radius decreases the maximum forming angle and increases the initial drawing angle. Higher sheet thickness was found to provide better formability. Elastic rebound increases with increasing initial pull angle and decreases with increasing thickness. From presented results it was revealed that PE and PA are suitable for incremental sheet forming due to their high elongation, while POM is not suitable for this purpose. The effect of PVC sheet thickness, feed rate, spindle speed, and other parameters have been experimentally investigated in [4] as well as the influence of the diameter of tool head on polymer incremental forming performance. An angle-varying cone was used in this experiment, and the largest forming limiting angle was taken as the experimental index. The obtained results have showed that the spindle speed and tool head diameter significantly influence the properties of the PVC sheet.

One of the key parameters of the SPIF process is the force developed by the tool which deforms the polymer sheet [5,6]. In paper [5], the dissipation of plastic energy during the SPIF process is divided into membrane, bending, and shear energy conditions. FEM modeling is used to analyze the sensitivity of deformation mechanisms to SPIF process variables. The results of the SPIF process and the quality of the products are related to formability and geometric accuracy by controlling the contribution of each deformation mode by adjusting the process variables. The paper [6] presents two axial force prediction procedures for polymer sheet SPIF. In particular, a numerical FEM model that takes into account the hyper elastic-plastic constitutional equation and a simple semi-analytical model that extends the known specific energy concept used in machining.

Despite advances in polymer sheet SPIF and constant attempts to produce durable products, the material can still fail [7,8,9]. The effect of changes in the polymer sheet SPIF process parameters and their interaction are described in paper [7]. This is due to the increased tendency to carve or file material. The high height of the steps reduces the formability and increases the tendency to wrinkle. The influence of product configuration and increased depth on failure modes during SPIF was investigated in paper [8]. Higher depths have been shown to increase polymer formability. It has been observed that the advantages of higher formability and shortened forming time at greater depths are limited by sheet wrinkling. The effect of tool speed and increased depth on the polymer tearing and wrinkling is investigated experimentally in paper [9]. Compared to the metal sheet SPIF, a higher incremental depth increases the formation of the polymer, but, as has been observed, this advantage is limited by the sheet wrinkling at high incremental depths and depends on the shape of the object to be formed.

In recent years, in prosthetic sector has been a growing interest in small-batch processes using biocompatible materials. SPIF technology can meet these requirements [10,11,12]. Sheets of polycaprolactone (PCL) and ultra-high molecular weight polyethylene (UHMWPE) were processed and shaped by the SPIF in the [10]. The performance of these biocompatible polymers in SPIF was assessed through the variation of four main parameters: the diameter and speed of the forming tool, the feed rate, and the step size based on a Box–Behnken design of experiments of four variables and three levels. It was found that during SPIF processing, it is key to reach the temperature of this transition for the forming of PCL and UHMWPE since it is associated with slippage between crystallites. The use of thermoplastic sheets as a raw material for SPIF is still not widespread, so the manufacturing of polymeric prostheses is also carried out using this technology. With this in mind, the aim of the article [11] is to obtain the real cranial geometry of a possible prosthesis produced in the SPIF process using a biocompatible polymer. The actual cranial fracture geometry is obtained from computed tomography and processed until a CAD model is obtained. From it, trajectories were defined and cranial geometry was made. In the work [12] SPIF experimental tests using two different biocompatible polymers have been carried out following a Box–Behnken design for four factors and a survival analysis. The maximum forming force, surface roughness and maximum forming displacement response achieved in the experiments has been statistically analyzed and empirical models for each material have been obtained.

The SPIF of multilayer polymeric and other materials is often limited compared to monolayer sheets [13,14,15,16]. An experimental study, presented in [13] was conducted to investigate the influence of process factors on the performance of composite materials. The thermoplastic matrix reinforced by 15% fiberglass SPIF was processed by adding an external heating source on the three-axis CNC milling machine. In paper [14], a new method for forming glass-fiber reinforced polymers has been proposed. This method uses a SPIF that assists in heating hot air and forms a thermoplastic matrix polyamide. A glass-fiber-reinforced polymer was placed between two layers of Teflon and of metal sheets. In paper [15] was investigated the polypropylene-based composites filled with different amounts of functionalized multi-walled carbon nanotubes (f-MWCNT) prepared using a melt mixing process and formed using the SPIF process. The paper [16] examines how additional depth, metal thickness, and polymer thickness affect the SPIF formability and failure modes of adhesive metal-polymer laminate sheets.

The aim of the study [17] is to investigate how the formability of thermoplastics has been improved using thermal-assisted SPIF. Hot air increased the temperature in the localized region in front of the tool. The SPIF molding device was modified by creating a specialized tool holder and nozzle that heats the polymer sheet to a temperature above room temperature but below the glass transition temperature and acts on the forming loads.

The analysis of the research results shows that there is not enough information about the most efficient polymer heating methods that ensure a fast and high-quality SPIF process. There is also a lack of research on the spontaneous deformations of the heated polymer sheet affecting the accuracy of the SPIF product. An innovative SPIF heating method for a PVC polymer sheet to maintain the stiffness of the polymer plate around the heated tool area and to eliminate the need for a backing plate is proposed in this article. The motivation of this research is related to the development of a new, inexpensive and hitherto rarely used technology due to unpredictable thermal deformations of polymer sheets. It is these deformations that partly determine the quality of the product being developed and the ability to control them would cause their spread in the industry by replacing the much more expensive 3D printing in the development of small-batch products.

## 2. Numerical Investigation of Polymer Sheet Forming Parameters

In this section the analytical research results of the sheet forming process parameters—heating temperature dependency from time and displacement of polymer sheet from gravity are presented. By using Ansys Transient Thermal together with Transient Structural analysis, a computational FE model of the PVC Trovidur ESA-D sheet, which geometric dimensions and properties of the elements used for the numerical calculations are collected in Table 1, was created.

Numerical analysis was conducted in two stages: in the first stage transient thermal analysis was carried out and thus temperature dissipation on the sheet polymer was obtained. During the second stage this dissipation of temperature was transferred to the transient structural analysis as the input parameter of the thermal load and after the analysis displacement of the polymer sheet from earth gravity was obtained. During the first stage of numerical analysis PVC sheet, which parameters are presented in Table 1, was excited with convection on the surface area of 25 mm offset from all edges of one surface, which corresponds area affected by thermal gun during experimental research. Ambient temperature and conditions were evaluated by adding another convection with different parameters on the rest of the surface of the sheet. A scheme of this stage of numerical analysis is presented in Figure 1a and computational model with boundary conditions are presented in Figure 1b. The finite element model (FEM) numerical simulation data of this stage are collected and presented in Table 2.

During both stages of the simulation the analysis time was divided into steps. The total time of simulation was 780 s. The time of the first stage was 5 s and this stage was divided into 50 sub steps. The second stage was 780 s and this stage was divided into 100 sub steps.

The simulation results are the temperature dissipation of the upper surface of the sheet, on which the ambient convection was added, so this surface is opposite to the heating surface, 2 s after the start of heating, shown in Figure 2a. Temperature dissipation of the same surface of the sheet after 780 s from the start of heating is presented in Figure 2b.

The simulation results allowed to determine the average and maximum dependence of the polymer sheet surface temperature on time (Figure 3), from which it can be seen that the sheet surface temperature changes nonlinearly over time until it settles.

Results of this simulation stage revealed that temperature distribution on the opposite to the heating surface is similar to heating area, what means that in polymer sheet SPIF process it is very important to carefully select geometry of the heating gun flow as well as of the surface of the polymer sheet. Modeling results presented in Figure 3 show that average temperature of the opposite to the heating surface is equal to 42.4 °C after 780 s from the start of heating and maximum is equal to 51.7 °C at the same instance. By using these results, it is possible to determine the required temperature and thus set the required time from the start of heating to begin SPIF process.

The next stage of simulation was carried out using these results as input loads for transient structural analysis. During this stage polymer sheet was fixed on the 25 mm offset from the sheet edge on the same side as heating gun and standard earth gravity vector was added perpendicularly to this surface. The calculation scheme of this stage of numerical analysis is presented in Figure 4a and computational model with boundary conditions in Figure 4b.

FE model numerical simulation data of this stage are presented in Table 3.

The simulation results of the total deformation of the polymer sheet after 780 s from the start of heating are presented in Figure 5a, and the equivalent Von Mises stress dependencies at the same instance are presented in Figure 5b.

The simulation results of the average and maximum deformation of the sheet under the earth gravity versus time are presented in Figure 6.

The simulation results revealed that maximum deformation of the polymer sheet from standard earth gravity exceeds 10.9 mm, while the average deformation of the polymer sheet was 2.6 mm. These results show the need to carefully select SPIF parameters such as initial and forming depth of the tool as well as required forming forces.

## 3. Validation of Numerical Simulation Results

The scheme of experimental set-up is presented in Figure 7a and the experimental set-up view is presented in Figure 7b. During the experimental research, PVC ESA-D polymer sheet (1) was fixed on the frame (2) by 25 mm offset area from its edge, like it was in the theoretical research. The polymer sheet (1) was heated by blowing hot air with blowing system CT-850D (Acifica, Inc., San Jose, CA, USA) (6) from lower surface through the hole formed in frame (2) and opposite surface displacement was measured with laser displacement meter Kyence LK-G82/3001 (Keyence Corporation, Neu-Isenburg, Germany) (3). Signal obtained from this meter was analyzed using digital oscilloscope PicoScope-6403 (Pico Technology Ltd., Cambridgeshire, UK) (5). At the same instance temperature of the polymer sheet surface was measured using thermal imaging camera FLIR T450sc (FLIR Systems Inc., Wilsonville, OR, USA) (7).

The experimental research results of the polymer sheet surface temperature versus time and polymer sheet displacement from gravity versus temperature are presented in Figure 8a,b, respectively. As it can be seen from the results presented in Figure 8a, maximum temperature of the polymer sheet after 2 s from the start of heating is equal to 25.8 °C and after 780 s is equal to 45.5 °C. Figure 8b shows that displacement of the sheet surface increase significantly after surface reaches temperature 31 °C and is maximum when temperature of the polymer sheet is 45.5 °C.

The obtained results from the experimental research are close to the results obtained with the numerical simulations presented in Figure 3 and Figure 6. Temperature rate at the initial zone is lower due to a smaller heating area of hot air blowing system compared to heating conditions used in FEM analysis. This revealed the possibility to use numerical research in the future to determine required parameters for SPIF avoiding experimental determination and thus avoiding expenses related to the manufacturing of the specimens.

## 4. Investigation of the Polymer Sheet Advanced Heating Device

In Figure 6 and Figure 8b, in order to eliminate the numerically obtained and experimentally validated self-deformations of the heated polymer sheet resulting from earth gravity, it is necessary to use backing plates to constrain these deformations. However, this way not only increases production costs but also slows down processes, as backing plates have to be adapted to the forming of specific products, which is undesirable for unit production. A much more efficient method is associated with the use of a tool that, in contact with the polymer sheet, also heats it to a controlled temperature. In addition, the heating of the polymer sheet is much faster. To confirm this idea, an additional set of numerical investigations, which are generally similar to those presented in Section 2, were carried out.

During this numerical simulation, the heating temperature dependency from time and displacement of polymer sheet from gravity are presented, the same as in Section 2. In this modeling, the heating is modeled as a point heating source, while in Section 2 it was modeled as air flow from a heat gun. By using ANSYS transient thermal together with transient structural analysis, a computational FE model of the same PVC Trovidur ESA-D sheet was created, and the geometric dimensions and properties of the elements used for the numerical calculations are collected in Table 1. Numerical analysis as described in Section 2 was conducted in two stages. The PVC sheet was excited with the heat flow source defined by the heat power varying in time, which corresponds to the point heating source. The properties of excitation heat flow parameters are presented in Table 4, and the corresponding curve in Figure 9.

The ambient temperature and conditions were evaluated the same as in Section 2, by adding another convection with different parameters on the rest of the surfaces of the sheet. A scheme of this stage of numerical analysis is presented in Figure 10a, and the computational model and boundary conditions are presented in Figure 10b. The FEM numerical simulation data together with the main excitation heat flow parameters of this stage are collected and presented in Table 5.

During both stages of the simulation the analysis time was divided into steps. The total time of simulation was 120 s. The time of the first stage was 3 s and this stage was divided into 50 sub steps. The second stage was 120 s, and this stage was divided into 100 sub steps.

The simulation results are the temperature dissipation of the lower surface of the sheet, i.e., opposite to the heat flow surface, 1.92 s after the start of heating, shown in Figure 11a. Temperature dissipation of the same surface of the sheet after 120 s from the start of heating is presented in Figure 11b.

The simulation results allowed determination of the average and maximum dependence of the polymer sheet surface temperature on time (Figure 12a), from which it can be seen that the sheet surface temperature changes nonlinearly over time until it settles.

The results of this simulation stage revealed that the difference between the maximum and average temperature of the PVC sheet is significantly higher compared to the results presented in Figure 3, which means that the heating occurs on quite a small zone of the surface. The obtained results showed that the average temperature of the measuring surface is equal to 22.1 °C after 120 s from the start of heating, and the maximum is equal to 49.3 °C at the same instance. The maximum temperature in the 120 s time period is 57.9 °C after 1.86 s from the beginning of heating. The shorter heating time and smaller heating area is more energetically effective comparing to heating by air gun. Additionally, by using these results, it is possible to determine the required temperature and thus set the required heating flow power for the SPIF process.

The next stage of the simulation was identical to that described in Section 2, and its scheme is the same as in Figure 4, and the main parameters were the same as in Table 3, and it only differed in simulation time, which was equal to 120 s. The simulation results of the total deformation of the polymer sheet after 120 s from the start of heating are presented in Figure 13a and the equivalent Von Mises stress dependencies of the polymer sheet at the same instance are in Figure 13b.

The simulation results of the average and maximum deformation of the sheet under the earth gravity versus time are presented in Figure 12b. The simulation results revealed that the maximum deformation of the polymer sheet from standard earth gravity exceeded 1.9 mm, while the average deformation of the polymer sheet was 0.42 mm. Both the maximum and average displacement of the sheet from gravity were significantly smaller compared to the results presented in Section 2. This revealed the possibility of using a point heating device for the SPIF process; hence, a novel forming tool with point-heating ability is required.

An image of the proposed advanced heating device, for which the description was applied for patenting by the authors [19], is presented in Figure 14.

At the contact zone with the polymer sheet, a tool tip is a free 3D rotating metal sphere (1) in the ring-shaped magnet (2), a waveguide (3) is placed in the housing of the tool and is excited by ultrasonic vibration transducer–piezoceramic discs (4), thus reducing the metal sphere friction with the ring-shape magnet and facilitating its free rotation. The metal sphere is heated by a laser beam which is directed at it via fiber optics (5), and the heating temperature is controlled by means of feedback via a temperature sensor (6).

The paper [20] explains the influence of mechanical ultrasonic vibrations generated by a piezoelectric actuator, significantly reducing the frictional forces in the contact area of metal parts due to the superposition of ultrasonic vibrations. A modification of the Coulomb‘s friction law can be applied to this kind of vibrating friction contact.

A schematics of the polymer sheet SPIF using a local heating tool and a general view of the equipment are shown in Figure 15.

The heated tool moving on the sheet surface locally increases the plasticity of the polymer in the contact zone with less deforming force. This method of heating does not reduce the stiffness of the polymer around the tool contact area, which eliminates the need for a backing plate. This allows the fabrication of products with sharper properties without the need for a supporting structure. In order to reduce the friction between the sphere and ring magnet, the tool housing is excited by ultrasonic vibrations with a piezoelectric transducer.

## 5. Experimental Investigation of Polymer Sheet SPIF Parameters

To verify numerical investigation results experimental research based on simulation parameters and process presented in Section 2 were carried out. During this phase of research experimental investigations of forming four different geometric shapes figures from polymer sheet, which material properties are presented in Table 1 was carried out. Scheme of experimental set-up which description was applied for patenting by the authors [21] and stand view are presented in Figure 16a,b, respectively. The main data of experimental research are collected and presented in Table 6.

During the experimental research, the forming tool (4) with a freely rotating tool on its end was used. Software with four different modifications of circle, square, star, and flower spatial geometric shapes for the robot IRB1200 (ABB Robotics & Discrete Automation, Västerås, Sweden) (3) was created with a PC (7). Polymer sheet (1) was fixtured on a frame (2) and heated with a Toolland PHG2 (Tooland Inc., San Carlos, CA, USA) hot air gun (5). The temperature within the required range based on the results presented in Section 2 and Section 3 was controlled using a FLIR T450sc (FLIR Systems Inc., Wilsonville, OR, USA) thermal imaging camera (6).

Figure 17 shows the transient state of the temperature distribution on the surface of the forming sheet when it is heated using hot air gun and advanced heating device, presented in Section 4. The lowest temperature values (40–60 °C) were measured near the tool-sheet interface and in the center of the work piece. Note that, during the forming process the temperature must not exceed the softening temperature of 75 °C.

Figure 18 illustrates incrementally formed polymer sheets of different spatial geometry with processing parameters used during experiment which are listed in Table 6.

## 6. Polymer Sheet Robotized SPIF Tests with Different Tools

A standard version of ABB controller IRC5 M2004 was used to control industrial robot IRB1200 M2004. Unfortunately, it does not have analog inputs. In order to extend capabilities of the controller IRCS M2001 a DeviceNet network adapter CREVIS NA-9111 with analog input channels extension PicoScope-3424 (Pico Technology Ltd., Cambridgeshire, UK) was used. The Mark-10 STJ100 (Mark-10 Corp., Copiague, NY, USA) torque sensor that was used in the experiment can provide analog output of ±1 V at full scale and extended analog input, with a dynamic range of 0–10 V (PicoScope-3424). Therefore, in order to take full advantage of the dynamic range of 12 bits analog input the signal was amplified five times at the same time providing a positive offset voltage. The schematics of a robotized step-by-step feedback system is presented in Figure 19.

Experimental thermoplastic forming tests were performed with 2 mm thick PVC Trovidur ESA-D polymer sheet. The geometry of the formed sample is cone-shaped, large diameter 140 mm, small diameter 80 mm, embossing depth 30 mm. The vertical and horizontal steps of the pressure are equal to 0.5 mm. The experimental tests were performed with three different forming tools:Forming tool with Ø17 mm freely rotating sphere.Forming tool with Ø10 mm rotating sphere, supported by a ring-shaped magnetic holder.Forming tool with Ø10 mm rotating sphere, supported by a ring-shaped magnetic holder and heated to 46 °C.

The results of the experimental study of the dependence of force on the forming tool and the depth of embossing are presented in Figure 20.

From the presented graphs, it can be seen that in the case of a point polymer heating by forming tool, the vertical component of the force acting on the polymer sheet is the smallest, which allows to expect the lowest probability of inducing forming defects.

In all polymer forming processes, the first step is ensuring the quality of the polymer materials, by means of controlling the material temperature. Upon heating, most polymers undergo thermal transitions that provide insight into their morphology. The strength of the tested polymer samples decreases with the increasing of heating temperature and accordingly the material becomes softer. With the SPIF, a polymer sheet is heated to a target temperature, then formed to a specific shape. A polymer sheet is locally deformed by a forming tool which moves on the surface of the polymer sheet. Since the moving tool is following a defined path in 3D space, this forming process is flexible to be applied to arbitrary shapes. A reasonable localized temperature should be produced below the glass transition temperature, which was ideal to soften the thermoplastic sheet and keep the stiffness at a proper level. A SPIF device is modified through the development of a specialized tool holder and nozzle which heats the polymer sheet to temperatures higher than the room temperature but below the glass transition temperature of the polymer and applies the forming loads. The present process provides localized and controlled heating through the contact of the sheet with a tool, and then constitutes a viable way to preserve the flexible nature of this technology. A preliminary experimental campaign consisted of dynamic mechanical analyses on the polymer sheets to get an idea of the influence of the tool shape on the sheet heating, in order to reach temperatures that soften the polymeric sheets and do not compromise their surface quality. The heated tool moving on the sheet surface locally increases the plasticity of the polymer in the contact zone with less deforming force does not reducing the stiffness of the polymer around the tool contact area and eliminating the need for a backing plate.

## 7. Conclusions

The analysis of the polymer sheet SPIF has showed that the main focus should be on the control of the heating temperature:The numerical study of the polymer sheet heating parameters was conducted by blowing it with a stream of hot air, performed using the ANSYS transient thermal and transient structural analysis, which revealed the temperature dependency from heating time and thus deformation time.The results showed that average temperature of the opposite to the heating surface is equal to 42.4 °C after 780 s from the start of heating and that the maximum is equal to 51.7 °C at the same instance. The maximum deformation of the polymer sheet from standard earth gravity exceeds 10.9 mm, while the average deformation of the sheet was 2.6 mm.An innovative solution was proposed to heat the polymer sheet at the point of contact with the forming tool using laser beam energy.The numerical simulation results of the point-heated device showed that the heating time to maximum temperature of 57.9 °C was 1.86 s. The maximum deformation from gravity and temperature did not exceed 1.9 mm, with an average of 0.42 mm during the 120 s time interval.The experimental studies of the robotized polymer sheet SPIF have shown that in the case of the proposed point heating method the forming force decreases, the heating time of the sheet decreases, and the forming process takes place without supporting the polymer sheet by backing plate.

## 8. Patents

There are two patents resulting from the work in this manuscript: “Sheet parts incremental forming device”, patent application no. LT2021 549, Lithuania patent bureau and “Incremental forming machine for sheet plastic parts”, patent application no. LT2020 528, Lithuania patent bureau.

## Figures and Tables

**Figure 1 sensors-21-07459-f001:**
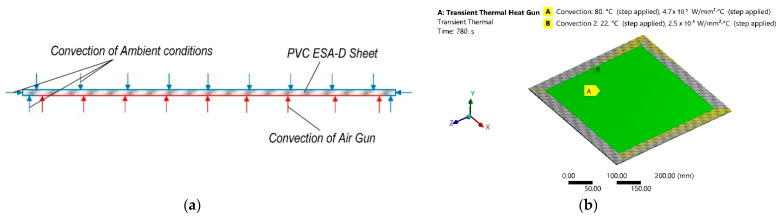
Transient thermal numerical analysis: (**a**) scheme; (**b**) computational model with boundary conditions.

**Figure 2 sensors-21-07459-f002:**
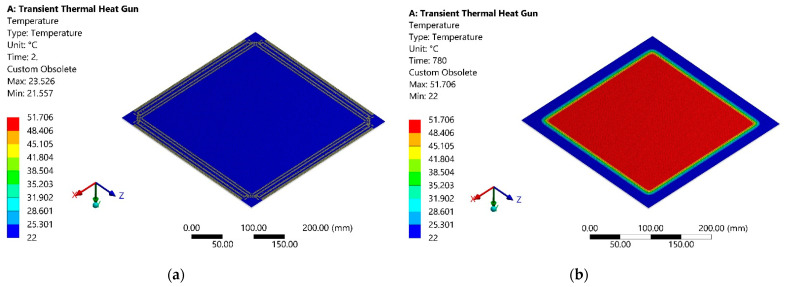
Temperature distribution on the opposite to the heating surface: (**a**) after 2 s from heating beginning; (**b**) after 780 s from the start of heating.

**Figure 3 sensors-21-07459-f003:**
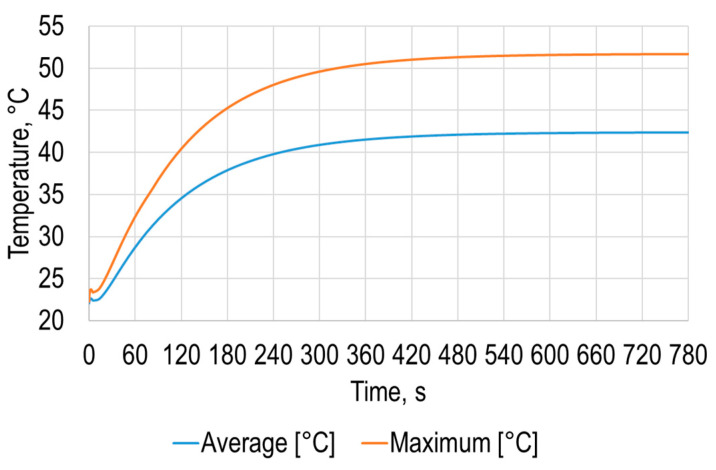
Average and maximum temperatures of the opposite to the heating polymer sheet surface.

**Figure 4 sensors-21-07459-f004:**
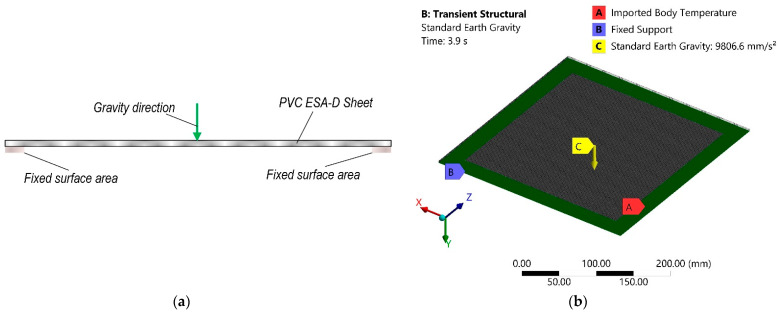
Transient structural numerical analysis: (**a**) scheme; (**b**) computational model with boundary conditions.

**Figure 5 sensors-21-07459-f005:**
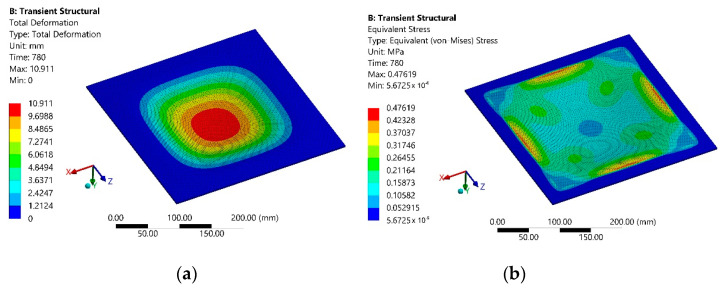
Modeling results of the second stage after 780 s from the heating beginning: (**a**) total deformation; (**b**) equivalent Von Mises stress.

**Figure 6 sensors-21-07459-f006:**
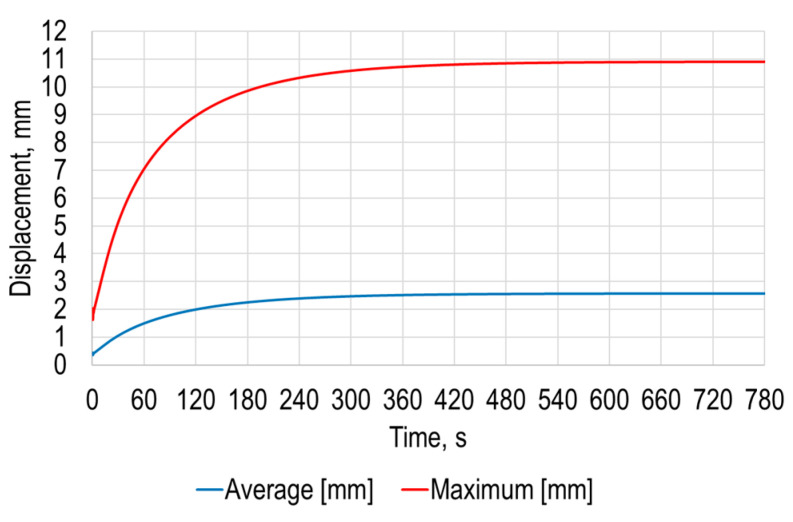
Average and maximum deformation under the earth gravity versus time of the polymer sheet.

**Figure 7 sensors-21-07459-f007:**
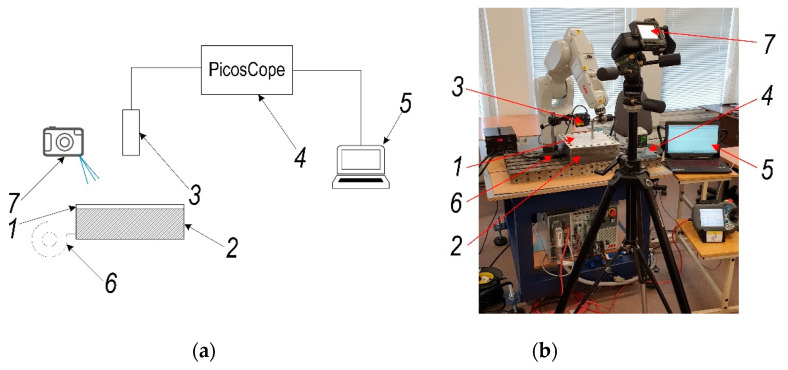
Experimental set-up for polymer sheet forming parameters validation: (**a**) scheme; (**b**) set-up view: 1—PVC ESA-D polymer sheet, 2—holding frame, 3—laser displacement meter Kyence LK-G82/3001 (Keyence Corporation, Neu-Isenburg, Germany), 4—digital oscilloscope PicoScope-6403 (Pico Technology Ltd., Cambridgeshire, UK), 5—PC, 6—hot air blowing system CT-850D (Acifica, Inc., San Jose, CA, USA), 7—thermal imaging camera FLIR T450sc (FLIR Systems Inc., Wilsonville, OR, USA).

**Figure 8 sensors-21-07459-f008:**
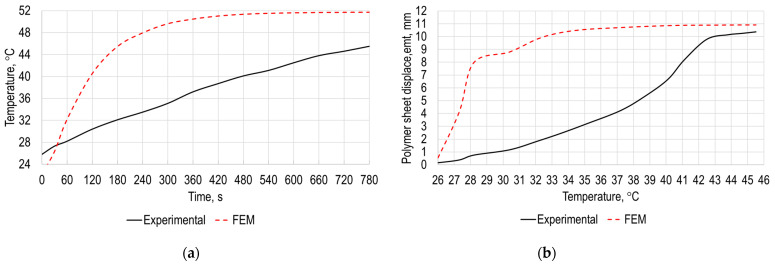
Experimental validation results: (**a**) polymer sheet temperature versus time; (**b**) polymer sheet displacement versus temperature.

**Figure 9 sensors-21-07459-f009:**
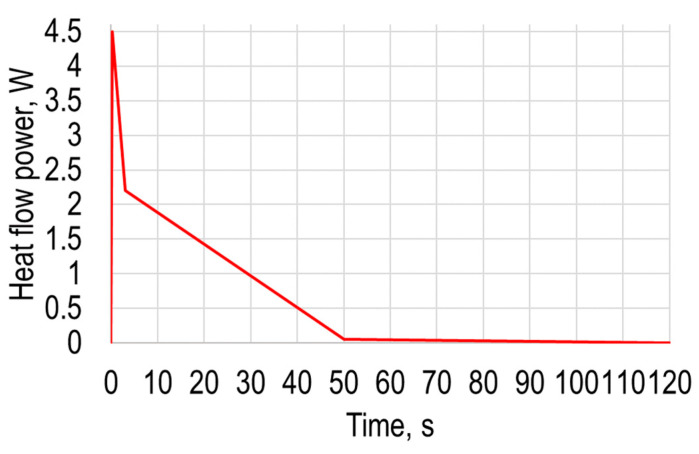
Excitation heat flow curve.

**Figure 10 sensors-21-07459-f010:**
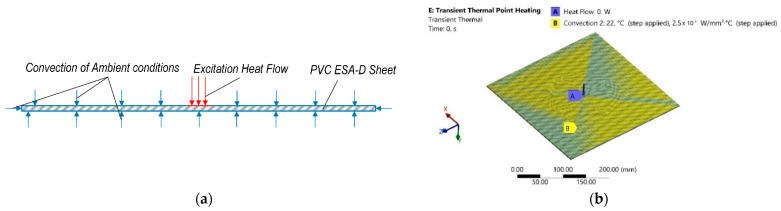
Transient thermal numerical analysis: (**a**) scheme; (**b**) computational model with boundary conditions.

**Figure 11 sensors-21-07459-f011:**
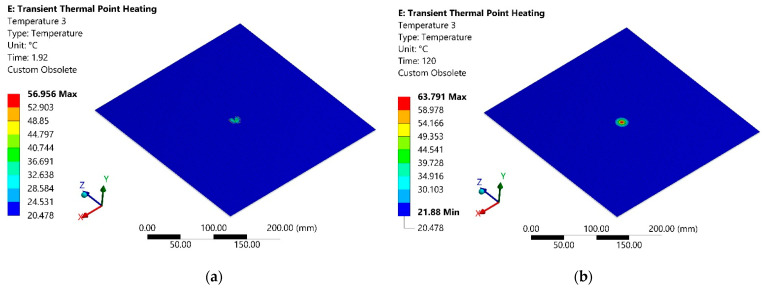
Temperature distribution on the opposite to the heating surface: (**a**) after 1.92 s from heating beginning; (**b**) after 120 s from the start of heating.

**Figure 12 sensors-21-07459-f012:**
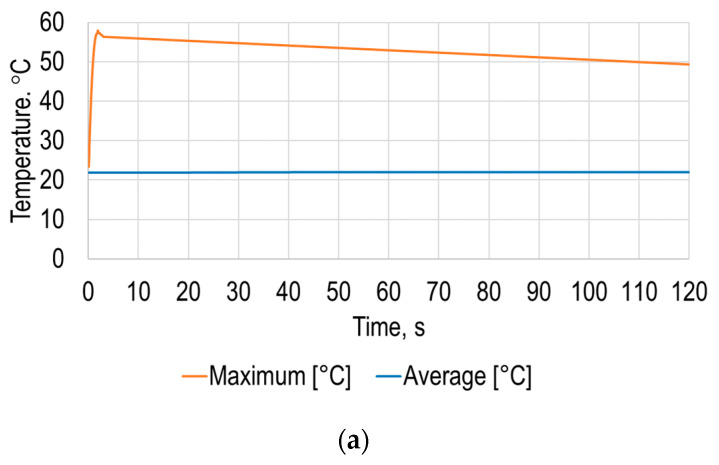
Modeling results: (**a**) average and maximum temperatures of the opposite to the heating polymer sheet surface; (**b**) average and maximum deformation under the earth gravity versus time of the polymer sheet.

**Figure 13 sensors-21-07459-f013:**
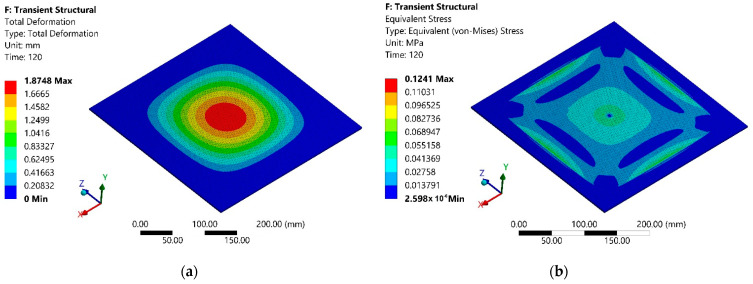
Modeling results of the second stage after 120s from the start of heating: (**a**) total deformation; (**b**) equivalent Von Mises stress.

**Figure 14 sensors-21-07459-f014:**
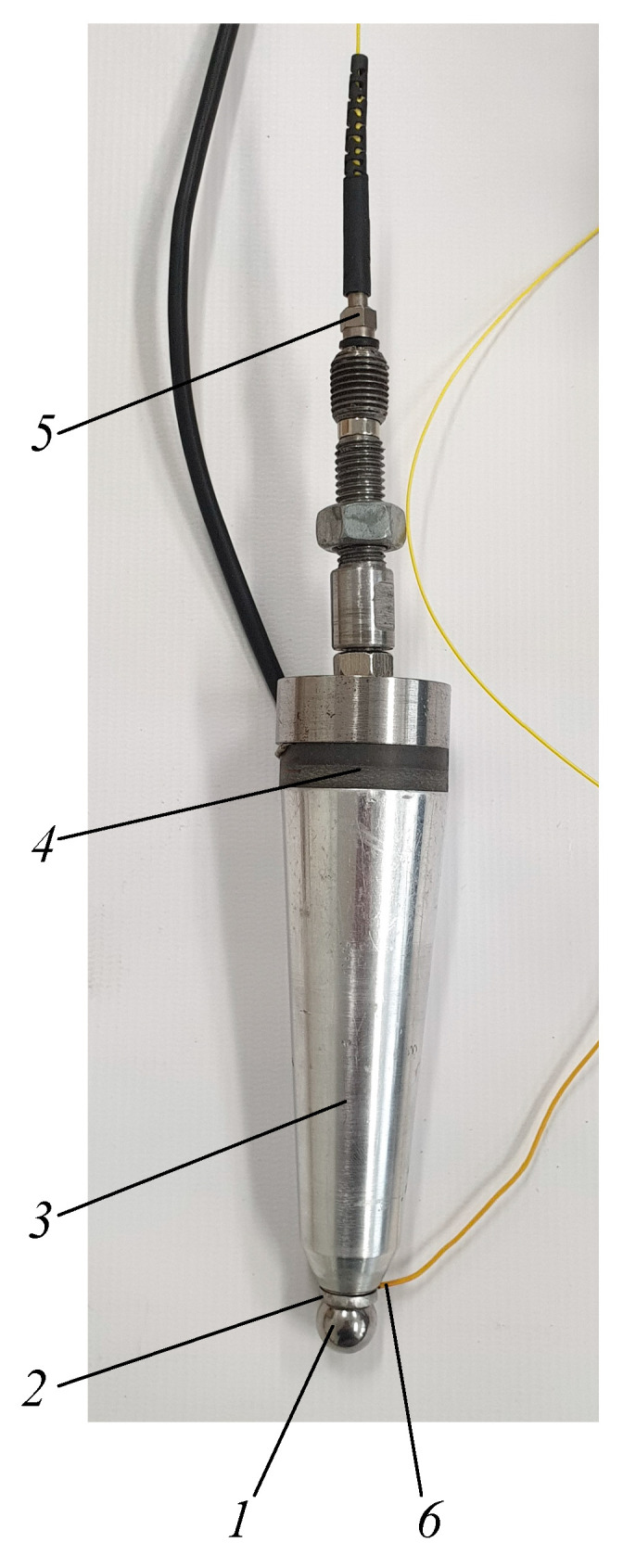
Polymer sheet SPIF tool: 1—metal sphere, 2—ring-shaped magnet, 3—waveguide, 4—ultrasonic vibration transducer-piezoceramic discs, 5—fiber optics, 6—temperature sensor.

**Figure 15 sensors-21-07459-f015:**
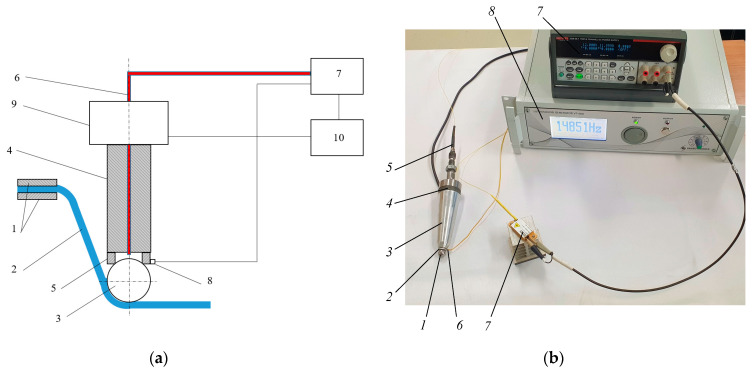
Polymer sheet SPIF with a laser heating: (**a**) schematics; (**b**) equipment: 1—metal sphere; 2—ring-shaped magnet; 3—waveguide; 4—ultrasonic vibration transducer-piezoceramic discs; 5—fiber optics; 6—temperature sensor, 7—laser diode module BMU25-940-01-R (Oclaro Inc., San Jose, CA, USA) with laser beam intensity controller Keithley 2230-30-1 (Keithley Instruments Inc., Cleveland, OH, USA), 8—ultrasonic vibration controller Sensotronica VT-400. (KTU, Kaunas, Lithuania).

**Figure 16 sensors-21-07459-f016:**
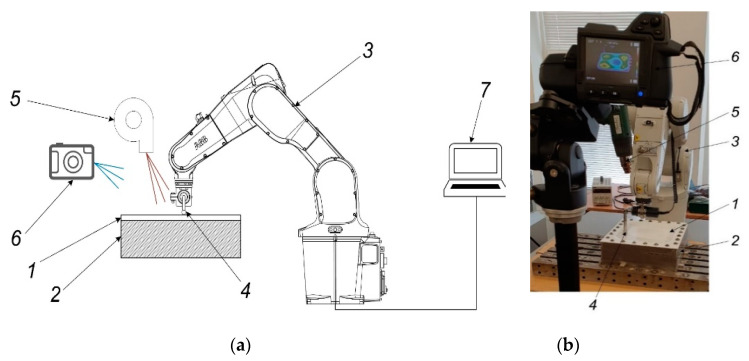
Experimental investigation of forming four geometric shapes from polymer sheet: (**a**) scheme; (**b**) stand view. Here: 1—PVC ESA-D polymer sheet, 2—holding frame, 3—robot ABB IRB1200 (ABB Robotics & Discrete Automation, Västerås, Sweden), 4—forming tool, 5—heat gun Toolland PHG2 (Tooland Inc., San Carlos, CA, USA), 6—thermal imaging camera FLIR T450sc (FLIR Systems Inc., Wilsonville, OR, USA), 7—PC.

**Figure 17 sensors-21-07459-f017:**
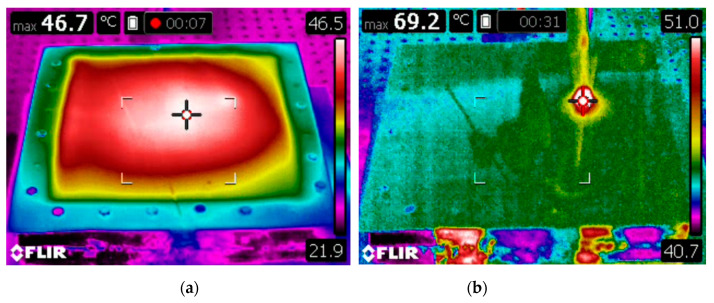
Thermal images of polymer sheet: (**a**) heated with air gun, (**b**) heated with advanced heating device.

**Figure 18 sensors-21-07459-f018:**
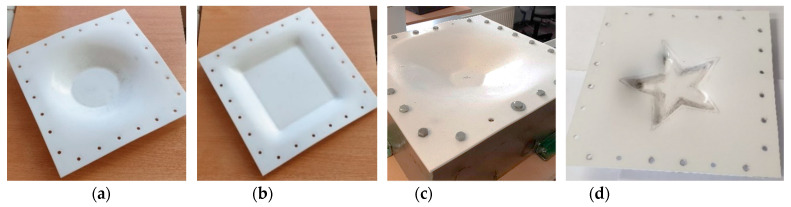
Photos of the incrementally formed polymer sheets of PVC ESA-D material was achieved by elevated temperature forming conditions: (**a**) spatial circular geometry; (**b**) spatial square geometry; (**c**) spatial flower geometry; (**d**) spatial star geometry.

**Figure 19 sensors-21-07459-f019:**
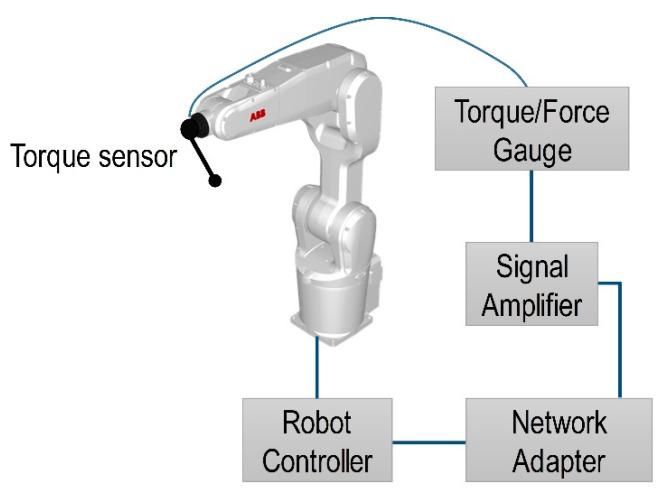
Schematics of a robotized polymer sheet SPIF step-by-step feedback system.

**Figure 20 sensors-21-07459-f020:**
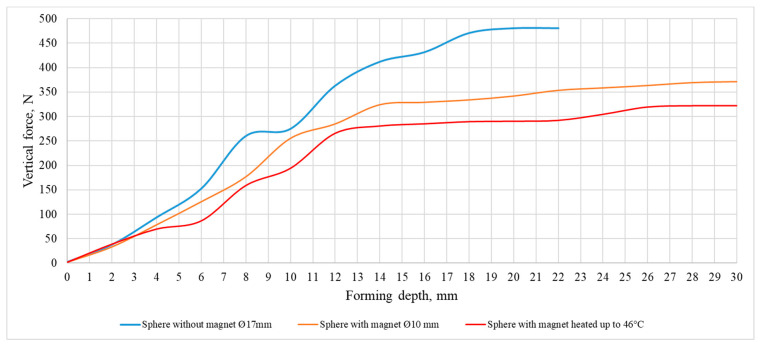
Dependence of the forming force on forming depth.

**Table 1 sensors-21-07459-t001:** Properties of the PVC Trovidur ESA-D material and geometric dimensions of the sheet used in calculation [18].

Parameter	Value	Unit
Length × width of the sheet	300 × 300	mm
Thickness of the sheet	3	mm
Density	1.41	g/cm^3^
Tensile stress at yield	47.75	N/mm^2^
Elongation at break	30.3	%
Modulus of elasticity in tension	2643	N/mm^2^
Notched Impact strength	9.09	mJ/mm^2^
Compressive strength	65.4	MPa
Vicat-softening temperature	75.0	°C
Coefficient of linear thermal expansion	70	10^−6^/K

**Table 2 sensors-21-07459-t002:** FE numerical simulation data of the first stage of analysis.

Parameter	Value	Unit
Mesh elements method	Hex Dominant	-
Number of finite elements	3600	-
Number of nodal points	25,803	-
Convection film coefficient of the heat gun surface area	47	W/m^2^
Convection temperature of the heat gun surface area	80	°C
Convection film coefficient of the rest ambient surface area	25	W/m^2^
Convection temperature of the rest ambient surface area	22	°C
Total time of calculation	780	s

**Table 3 sensors-21-07459-t003:** FE numerical simulation data of the second stage of analysis.

Parameter	Value	Unit
Mesh elements method	Hex Dominant	-
Number of finite elements	3600	-
Number of nodal points	25,803	-
Input load	Temperature	°C
Acceleration of gravity	9806.6	mm/s^2^
Total time of calculation	780	s

**Table 4 sensors-21-07459-t004:** Excitation heat flow parameters.

Time, s	Heat Flow Power, W
0	0
0.18	4.5
3	2.2
50	0.05
120	0

**Table 5 sensors-21-07459-t005:** FE numerical simulation data of the first stage of analysis.

Parameter	Value	Unit
Mesh elements method	Hex Dominant	-
Number of finite elements	15,124	-
Number of nodal points	103,231	-
Heat flow application geometry	Ø10 mm circle	^-^
Heat flow magnitude	Tabular (see Table 4)	W
Convection film coefficient of the rest ambient surface area	25	W/m^2^
Convection temperature of the rest ambient surface area	22	°C
Total time of calculation	120	s

**Table 6 sensors-21-07459-t006:** Experimental research data for incremental polymer sheet forming.

Parameter	Value	Unit
Radius of the forming tool sphere	8.5	mm
Step down	0.5	mm
Radial step	0.5	mm
Total forming depth	30	mm
Feed rate	100	mm/s
Major diameter of the geometric figure	150	mm
Minor diameter of the geometric figure	90	mm
Minimum temperature of the surface	40	°C
Maximum temperature of the surface	60	°C

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
