# Peer review of "Investigation of Advanced Robotized Polymer Sheet Incremental Forming Process"

_sensors, 2021, doi:10.3390/s21227459_

Round 1

Reviewer 1 Report

This paper investigated the advanced robotized polymer sheet incremental forming process. There are some issues, which need to be addressed. The detailed comments are given as follows.

  1. For the Section of Keywords, each keyword is suggested to be capitalized.
  2. The literature review needs to be concise.
  3. The motivation of this study is not clear. More paragraphs are suggested to be added to talk about the motivation.
  4. The new worthy gap and work on the furtherance of knowledge are suggested to be stated. Please clarify the innovation of this research.
  5. Some figures are suggested to be improved. For instance, the labels in Figure 1, Figure 3, Figure 4, Figure 5, Figure 7, Figure 10, Figure 11, Figure 12, and Figure 13. There are some noises in the labels of Figure 19.
  6. The deep discussions of the results are suggested to be added.
  7. It can be seen in Figure 20 that the changes of forming forces after a certain forming depth (e.g. 18 mm) become smaller. Why does this happen?
  8. The conclusions need to be improved. The major conclusions are suggested to be listed one by one.

Author Response

Dear reviewer, thank you for your time and quality evaluation of the publication. We are sending answers to your questions.

Reviewer 2 Report

The intention to enhance the stiffness of the polymer around the tool contact area and eliminating the need for a backing plate which determined the main question addressed is interesting in the moment work. But it seems the logic doesn’t match each other among the title, the abstract and body-paper, it will be very appreciated if the authors could re-consider and re-design.

As mentioned in the abstract,

1) why and how does the friction of which with the tool holder is reduced by ultrasound, and it should be discussed and analysed.

2) The effect of the free 3D rotating ball derived improving the surface quality should be investigated.

Author Response

(The authors gave the same response as above.)

Reviewer 3 Report

The paper presents a robotized system for the incremental forming of polymer sheets. In my opinion, the work can be improved with some changes.

  • The material of the sheets used in this work has to be indicated both in the abstract and in the introduction section
  • Figure 8. It can be useful to report on these figures the overlapping of the FEM results
  • Figure 8b. It is not clear how it has been designed. In fact, it results that the displacement tends to zero at higher temperatures
  • I think that in Figures like Figure 2 and 11 it is more correct to use the term "distribution" instead of "dissipation" when you talk about the temperatures
  • It is not clear the usefulness of carrying out the geometries reported in Figure 18, since you do not report a comparison with geometries realized with different tools or you do not report an evaluation of features like geometrical characteristics or forces
  • Figure 19 has to be corrected since it reports underlined words
  • Table 7 can be eliminated since Figure 20 reports the same information. Moreover, it is necessary to explain why the curve obtained for the "sphere without magnet" case ends first.

Author Response

(The authors gave the same response as above.)

Round 2

Reviewer 2 Report

Dear authors,

1) As mentioned, "ultrasonic friction reduction is well known in metal-metal contacts", please explain the exact mechanism. It will be appreciated if 1 or more related reference could be provided.

2) Due to the vibration, "the stick phase in the contact phase vanishes and only sliding occurs", please explain the mechanism.

Author Response

Dear reviewer, thank you for your time and quality evaluation of the publication. Please find attached answers to your questions. 
